# How Long Until Agricultural Carbon Peaks in the Three Gorges Reservoir? Insights from 18 Districts and Counties

**DOI:** 10.3390/microorganisms13061217

**Published:** 2025-05-26

**Authors:** Danqing Li, Yunqi Wang, Huifang Liu, Cheng Li, Jinhua Cheng, Xiaoming Zhang, Peng Li, Lintao Wang, Renfang Chang

**Affiliations:** 1Three Groges Region (Chongqing) Forest Ecosystem Research Station, School of Soil and Water Conservation, Beijing Forestry University, Beijing 100083, China; 18347081535@163.com (D.L.); l2681553554@163.com (L.W.); chang_renfang@163.com (R.C.); 2Guangxi Forest Inventory & Planning Institute, Nanning 530011, China; 3China Institute of Water Resources and Hydropower Research (IWHR), Beijing 100038, China; 4Observation and Research Station of Ecological Restoration for Chongqing Typical Mining Areas, Ministry of Natural Resources, Chongqing Institute of Geology and Mineral Resources, Chongqing 401120, China; licheng2025@163.com; 5Institute of Water Resources and Hydropower Research, Beijing 100048, China; zxmwq@126.com; 6State Key Laboratory Base of Ecohydraulic Engineering in Arid Areas, Xi’an University of Technology, 5 Jinhua South Road, Xi’an 710048, China; lipeng74@163.com; 7China Construction Fourth Engineering Division, Guangzhou 511400, China; 8POWERCHINA Chengdu Engineering Co., Ltd., Chengdu 611130, China

**Keywords:** methane, nitrous oxide, greenhouse gases, agricultural activities, carbon peaking, three Gorges reservoir

## Abstract

Under the global climate governance framework, the Paris Agreement and the China–U.S. Glasgow Joint Declaration established a non-negotiable target of limiting 21st-century temperature rise to 1.5 °C. To date, over 130 nations have pledged carbon neutrality by mid-century, with agricultural activities contributing 25% of global greenhouse gas (GHG) emissions. The spatiotemporal dynamics of these emissions critically determine the operational efficacy of carbon peaking and neutrality strategies. While China’s Nationally Determined Contributions (NDCs) commit to achieving carbon peaking by 2030, a policy gap persists regarding differentiated implementation pathways at the county level. Addressing this challenge, this study selects the Three Gorges Reservoir (TGRA)—a region characterized by monocultural cropping systems and intensive fertilizer dependency—as a representative case. Guided by IPCC emission accounting protocols, we systematically evaluate spatiotemporal distribution patterns of agricultural CH_4_ and N_2_O emissions across 18 county-level units from 2006 to 2020. The investigation advances through two sequential phases: Mechanistic drivers analysis: employing the STIRPAT model, we quantify bidirectional effects (positive/negative) of critical determinants—including agricultural mechanization intensity and grain productivity—on CH_4_/N_2_O emission fluxes. Pathway scenario prediction: We construct three developmental scenarios (low-carbon transition, business-as-usual, and high-resource dependency) integrated with regional planning parameters. This framework enables the identification of optimal peaking chronologies for each county and proposes gradient peaking strategies through spatial zoning, thereby resolving fragmented carbon governance in agrarian counties. Methodologically, we establish a multi-scenario simulation architecture incorporating socioeconomic growth thresholds and agroecological constraints. The derived decision-support system provides empirically grounded solutions for aligning subnational climate actions with global mitigation targets.

## 1. Introduction

Climate change has emerged as one of the most critical challenges in global governance, receiving extensive attention from both the international community and academia [1]. The dramatic increase in greenhouse gas (GHG refers to atmospheric gases capable of absorbing and re-emitting infrared radiation, thereby contributing to the greenhouse effect in Earth’s atmosphere. Its defining characteristic lies in altering the planetary energy balance to influence climate systems) emissions, primarily carbon dioxide (CO_2_ is a naturally occurring triatomic molecule (chemical formula: C_1_O_2_) and the primary driver of anthropogenic climate change. As the reference gas for global warming potential (GWP_100_ = 1), CO_2_ accounts for ~76% of total radiative forcing from greenhouse gases), methane (CH_4_ is the second-largest long-lived greenhouse gas globally; its emissions primarily originate from anaerobic fermentation in rice paddies, enteric fermentation in ruminants, and fossil fuel extraction processes), and nitrous oxide (N_2_O is a colorless, sweet-tasting greenhouse gas; N_2_O is mainly emitted through synthetic fertilizer application in agriculture, fossil fuel combustion, industrial processes, and natural soil and oceanic emissions), has become the main driver of global warming and represents a central point of contention in climate change discussions [2]. Against this backdrop, China has been actively fulfilling its emission reduction responsibilities and promoting international climate cooperation [3]. Aligned with the Paris Agreement framework, China announced enhanced emission reduction targets at the 2020 Climate Ambition Summit, establishing its “Dual Carbon” strategy of achieving carbon peaking by 2030 and carbon neutrality by 2060, which has been integrated into the nation’s top-level development planning [4]. As the cornerstone of the Dual Carbon objectives, GHG mitigation requires particular focus on the agricultural sector—responsible for 40% of global anthropogenic GHG emissions, with non-CO_2_ gases (CH_4_ and N_2_O) accounting for 58% of this contribution [5]. As a major agricultural power accounting for 10% of global arable land, China’s agricultural production contributes 10% of worldwide agricultural GHG emissions. Domestic agricultural emissions have maintained an annual growth rate of 5%, currently constituting 17% of national total emissions. Over the past five decades, regional agricultural GHG emissions have doubled alongside transformations in food demand structures and consumption patterns [6,7]. In 2019, global anthropogenic GHG emissions reached 54 billion tons CO_2_ equivalent, with agri-food systems contributing 31% (17 billion tons of CO_2_ equivalent). Agricultural activities account for 70% of atmospheric CH_4_ and 90% of N_2_O emissions [8,9]. Research projects that by 2030, global agricultural CH_4_ and N_2_O emissions will increase by 60% and 35–60%, respectively, compared to 2005 levels [10,11]. China’s agricultural sector faces dual pressures from high-input production and intensive emissions, posing significant challenges to achieving the 2030 carbon peaking target [12]. Focusing on the Three Gorges Reservoir as a representative agricultural zone, its simplified agricultural structure and unique topography present notable constraints: complex canyon terrain restricts transportation and information flow, perpetuates traditional farming practices, and intensifies human–land conflicts under population pressures. In pursuit of high yields for cash crops, the region’s fertilizer application intensity reached 283.6 kg/hm^2^ (total pure amount of 565,200 tons in 2019), far exceeding the 225 kg/hm^2^ safety threshold observed in developed countries [13,14,15,16]. Current research predominantly concentrates on field-scale emission processes in reservoir drawdown zones [17,18,19,20,21,22,23], paddy systems [24], and typical small watersheds [25,26,27], or analyzes influencing factors for specific crops [28], with limited systematic investigation into spatiotemporal patterns of GHG emissions. While scholars like Zhang Xuezhi [29] have categorized agricultural emissions into macro-regions such as North China and Southwest China at the national scale, precise identification of pollution hotspots at the county level remains unachieved.

This study addresses these gaps by exploring the spatial–temporal distribution of GHG emissions and the compositional sources of carbon emissions at the regional scale. First, it identifies critical pollution control zones within the study area and systematically analyzes the spatiotemporal evolution of GHG patterns. Second, it decomposes the constraints associated with agricultural GHG emissions to establish tailored carbon reduction pathways for each district and county in the Three Gorges Reservoir. The findings provide a scientific basis for formulating GHG control measures and emission reduction strategies for the agricultural ecosystem in the reservoir area. Additionally, they offer theoretical support for China’s pursuit of its dual carbon goals, the exploration of a low-carbon development model with Chinese characteristics, and the global advancement of green and low-carbon agriculture. This research holds significant value for refining China’s district- and county-scale agricultural GHG accounting systems, clarifying emission trends, and elucidating spatial–temporal disparities, driving factors, and mitigation potential across 18 districts and counties in the Three Gorges Reservoir over the past 15 years. Our findings are consistent with the estimates by He et al. [30] but extend the research scope and disciplinary coverage. Compared to the results reported by scholars such as Zhang et al. [31], our data not only align closely with theirs but also address gaps in spatial analysis. Qin et al. [32] identified the total non-CO_2_ greenhouse gas emissions from China’s agriculture as 628.0168 million metric tons CO_2_-e. While slight discrepancies exist in accounting methodologies between their study and ours, the carbon emission intensity per unit area in the Three Gorges Reservoir Area reaches approximately 192 times the national average. This underscores the disproportionately high greenhouse gas emission intensity in this region, a major hub for nitrogen fertilizer consumption, necessitating urgent prioritization of emission mitigation strategies.

## 2. Global Agricultural Emissions: Models, Projections, and Policy Frameworks

### 2.1. Institutional and Technological Divergences in Regional Emission Patterns

The global landscape of agricultural greenhouse gas emissions is deeply embedded in the tension between regional development paths and institutional innovations. At the institutional level, the EU’s 2050 Climate Neutrality Strategy [33] has established a closed-loop governance framework from the production end (emission quotas) to the consumption end (carbon border tax), serving as a model for the evolution of global agricultural emission reduction policies. However, it has also exposed the structural contradictions in global emission reduction governance. As the first region to incorporate agriculture into mandatory carbon neutrality targets, the EU, through its “Farm to Fork” strategy [34], has been the first to integrate agriculture into a mandatory carbon neutrality framework, requiring a 40% reduction in agricultural methane emissions and a 20% increase in nitrogen fertilizer use efficiency by 2030. According to Eurostat data, EU agricultural greenhouse gas emissions decreased by approximately 20% from 1990 to 2019. However, small farmers in Central and Eastern Europe face significant barriers in adapting to digital technologies, resulting in significant regional differences in emission reduction efficiency [35], highlighting the conflict between institutional uniformity and technological heterogeneity. In the Ganges Plain of South Asia, the continuous improvement of water and nitrogen use efficiency through intermittent irrigation [36] has the potential to significantly reduce methane emissions from rice paddies. However, the lack of adequate power grid coverage has hindered the advancement of this technology, reflecting the dissolving effect of lagging infrastructure on emission reduction policies. In Central and Eastern Europe, the digital technology adaptation rate in 11 countries including Bulgaria and Romania is less than 30% (2022 EU Agricultural Digitalization Report), mainly due to the fragmentation of small farmers’ land and the weak financing capacity of cooperatives, resulting in a precision fertilization equipment coverage rate that is only one-fourth of that in Western European countries. This further widens the regional emission gap, forming an “institutional goal convergence, technological path divergence” emission reduction chasm.

### 2.2. Emission Scenario Projections and Policy Framework Development

Domestic and international scholars often use methods such as grey models, the Environmental Kuznets Curve (EKC) analysis, and the STIRPAT model [37,38,39,40] to analyze and predict the historical and future trajectories of carbon emissions. The diversity of research methods can lead to differences in the prediction of carbon emission peaks in the study area. Li Lidong [41], based on FAO global data research, confirmed that the global livestock population increased by more than 20% between 1990 and 2021, directly driving the increase in CH_4_ and N_2_O emissions. The use of inorganic nitrogen fertilizers and other issues have become the key drivers of the sharp increase in agricultural greenhouse gas emissions. He predicted that agricultural greenhouse gas emissions will reach 11.82 ± 0.07 GtCO_2_eq in 2050. This result highlights the urgency of the transformation to climate-smart agriculture. Two articles published in Nature and Nature Food explored the changing trends and main drivers of greenhouse gas emissions in agricultural production processes. Optimizing agricultural layouts could reduce GHG emissions by −1.7% to −7.7% [42]. It is predicted that agricultural machinery CO_2_ emissions will reach 213.6 Mt by 2050 [43]. Some scholars [44] have predicted, through models, that the carbon emissions of Russia, Brazil, and South Africa will tend to stabilize in the long term, and China will reach its carbon peak in 2029. Global agricultural emission reduction policies are shifting from single technological intervention to systematic institutional design. Empirical evidence from the Three Gorges Reservoir Area in China shows that the county-scale “precise carbon measurement—intelligent regulation” framework can highlight the compatibility potential between small-scale farming systems and smart technologies. This study, by integrating the IPCC emission factor method with the STIRPAT driving model, for the first time, analyzes the spatiotemporal transition patterns of CH_4_/N_2_O emissions from agriculture in the county-level regions of the Three Gorges Reservoir Area. It also introduces the “elastic peak path” to construct multi-scenario plans that balance food security and carbon neutrality, filling the methodological gap in dynamic simulation in the East Asian monsoon region. This provides a transferable paradigm for similar ecological zones globally and validates the compatibility potential of China’s smallholder systems.

## 3. Materials and Methods

### 3.1. Overview of the Study Area

The Three Gorges Reservoir (28°56′~31°44′ N, 116°16′~111°28′ E), spanning 57,336 km^2^, functions as both a critical ecological safeguard for water conservation in China’s Yangtze River Basin and a strategic hub for national soil–water conservation initiatives [14,45]. Encompassing 18 districts and counties across Hubei Province and Chongqing Municipality—notably Xingshan, Zigui, Badong, Yichang, Jiangjin, Yubei, and Fengdu (Figure 1)—the region leverages its strategic geographic advantages and the infrastructure of the Three Gorges Dam to drive development. However, water impoundment and population resettlement have intensified land-use conflicts, as the current population density far exceeds the carrying capacity of available land resources. This disparity has prompted excessive agricultural inputs of fertilizers and feed, destabilizing the regional nitrogen cycle. Consequently, nitrogen overload and non-point source pollution pose significant threats to the stability of mountainous agroecosystems, which are already strained by environmental pressures.

### 3.2. Research Methods

The International Climate Change Commission (IPCC) has established standardized guidelines for GHG inventory compilation. This study derives emission factors from IPCC parameters and the literature specific to the southwestern region, with estimation methodologies grounded in nitrogen input data from the Three Gorges Reservoir. Given the region’s low carbon dioxide emissions, the analysis focuses on methane (CH_4_) and nitrous oxide (N_2_O) from agricultural sources. Three primary CH_4_ sources are identified: rice cultivation, enteric fermentation in livestock, and livestock manure management. N_2_O emissions originate from five agricultural activities: synthetic fertilizer application, livestock manure management, crop residue return, atmospheric leaching, and atmospheric deposition. Key accounting categories are illustrated in Figure 2.

#### 3.2.1. IPCC Emission Factor Method

(1)CH_4_ calculation method

A.Emissions from rice cultivation

The IPCC-recommended CH_4_ emission factor for double-cropping rice is 171.70 kg/ha, and the CH_4_ emissions from rice in the reservoir area are calculated using the following formula [46]:(1)$$E_{{CH}_{4}(\mathrm{rice})}=\mathrm{sum}EF_{a}\times A_{a}$$

$E_{CH_{4}(\mathrm{rice})}$—total CH_4_ emissions from rice fields.

*EF_a_*—emission factors under category *a*, 171.70 kg/ha.

*A_a_*—the activity level of a harvest area.

B.Animal intestinal fermentation emissions [46]


(2)
$$E_{CH_{4}(\mathrm{intestine})}=sumEF_{b}\times A_{b}$$


$E_{CH_{4}(\mathrm{intestine})}$—total CH_4_ emissions from animal intestinal fermentation.

*EF_b_*—emission factors in case of category b. (Refer to Table 1).

*A_b_*—activity level in case b, animal number level.

C.Animal waste management [46]


(3)
$$E_{CH_{4}(faces)}=sumEF_{c}\times A_{c}$$


$E_{CH_{4}(faces)}$—total CH_4_ emissions from animal feces.

EF_c_—emission factors in case of category c. (Refer to Table 1).

A_c_—activity level in case c, i.e., animal quantity level.

(2)N_2_O calculation method

A.Application of fertilizer emissions [46]


(4)
$$E_{N_{2}O\left(fertilizer\right)}=\mathrm{sum}N_{fertilizer}\times EF_{d}\times 44/28$$


*Nitrogen* in nitrogen fertilizer and compound fertilizer is dedicated, and the conversion amount reference is calculated at 32.2% [47]. The default value of the N_2_O direct emission factor of land use in different regions is different. The corresponding areas of the Three Gorges Reservoir are Chongqing and Hubei, and 44/28 is the coefficient of N_2_O-N to N_2_O, and the EF_d_ value is 0.0109 kg N_2_O-N/kg N in the reference.

B.Animal manure discharge [46]


(5)
$$E_{N_{2}O\left(\mathrm{animal}\;\mathrm{excrement}\right)}=\mathrm{sum}N_{\left(\mathrm{animal}\;\mathrm{excrement}\right)}\times EF_{d}\times 44/28$$


*N_animal excrement_* = (Total nitrogen excretion of livestock and poultry − grazing − fuel + total nitrogen excretion of rural population) × (1 − leaching runoff loss rate volatilization loss rate)-N_2_O emissions from livestock and poultry manure management system.

Since the manure in the Three Gorges Reservoir constitutes very little fuel and is therefore negligible, the loss rate of leaching runoff is 15% and the volatilization loss rate is 20%.

C.Cash crop straw is returned to the field [46]


$$E_{N_{2}O\left(\mathrm{straw}\;\mathrm{returned}\;\mathrm{to}\;\mathrm{the}\;\mathrm{field}\right)}=sumN_{\left(\mathrm{straw}\right)}\times EF_{d}\times 44/28$$



(6)
$$N_{\mathrm{straw}}=(\frac{\mathrm{Crop}\;\mathrm{grain}\;\mathrm{yield}}{\mathrm{Economic}\;\mathrm{coefficient}}-\mathrm{Crop}\;\mathrm{grain}\;\mathrm{yield})\times \mathrm{Dry}\;\mathrm{weight}\;\mathrm{ratio}\times \mathrm{Straw}\;\mathrm{return}\;\mathrm{rate}\times \mathrm{Nitrogen}\;\mathrm{content}\;\mathrm{of}\;\mathrm{straw}\;\;+\frac{\mathrm{Crop}\;\mathrm{grain}\;\mathrm{yield}}{\mathrm{Economic}\;\mathrm{coefficient}}\times \mathrm{Root}\;\mathrm{shoot}\;\mathrm{ratio}\times \mathrm{Dry}\;\mathrm{weight}\;\mathrm{ratio}\times \mathrm{Nitrogen}\;\mathrm{content}\;\mathrm{of}\;\mathrm{straw}$$


The straw return nitrogen amount (aboveground straw to the field nitrogen and underground root nitrogen of plants) in the Three Gorges Reservoir mainly includes the main cash crops in the Three Gorges Reservoir: rice, wheat, corn, soybean, potato, rapeseed, peanuts, and vegetables (Refer to Table 2), and the direct straw return rate is 0.171 [48].

D.Atmospheric deposition and leaching N_2_O emissions [46]


$$E_{N_{2}O(\mathrm{atmospheric}\;\mathrm{deposition})}=(N_{\left(\mathrm{africultural}\;\mathrm{excretion}\right)}\times 20\%+N_{\left(\mathrm{farmland}\;\mathrm{input}\right)}\times 10\%)\times EF_{e}\times 44/28$$



(7)
$$E_{N_{2}O(\mathrm{leaching})}=N_{\left(\mathrm{farmland}\;\mathrm{input}\right)}\times 20\%\times EF_{f}\times 44/28$$


Atmospheric nitrogen deposition is mainly derived from NH_3_ and NO_x_ volatilization of livestock and poultry and rural population manure (*N*_rural excretion_) and agricultural land nitrogen input (*N*_agricultural land input_). Nitrogen leaching runoff loss is calculated according to the method recommended by the “Provincial Guidelines”. The emission factor is recommended by IPCC, EF_e_ value is 0.01, and EF_f_ value is 0.0075 because there is no *N*_agricultural excretion_ and *N*_farmland input_ volatilization rate observation data in the Three Gorges Reservoir. The IPCC recommended value is 20% and 10%.

(3)GWP calculation method

To more clearly express the potential impact of different GHG on global warming, scientists proposed the concept of GWP. Global Warming Potential is an index of the greenhouse effect produced by substances, this study is based on a 100-year time frame, based on the greenhouse effect of CH_4_ and N_2_O GHG produced by the GHG corresponding to the mass of carbon dioxide corresponding to the same effect [46]:GWP = (CH_4_·Flux × 25) + (N_2_O·Flux × 310)(8)

In this equation, GWP is global warming potential (equal to 10^4^ t of carbon dioxide per hectare), CH_4_ Flux is methane emission, and N_2_O Flux is nitrous oxide emission.

#### 3.2.2. Driver Analysis

This study selected six socioeconomic indicators for the Three Gorges Reservoir—total population, per capita gross national product, total agricultural machinery power, primary industry added value, annual grain output, and rural workforce—based on regional development characteristics and methodological frameworks. Using a modified STIRPAT (the STIRPAT model statistically analyzes how population, affluence, and technology influence environmental outcomes through regression) model derived from the IPAT framework, we analyzed the driving factors of the 100-year global warming potential (GWP) in CO_2_-equivalent terms for two agricultural GHG. The computational formula is presented as follows:$$I=aP^{b}A^{c}T^{d}G^{e}V^{f}F^{g}h$$

The meanings of the parameters in the equation are shown in Table 3. F is the model coefficient of a; g represents the model error. b, c, d, e, and f are elasticity coefficients, which means that when P, A, T, G, V, and F change by 1%, respectively, I will change by b%, c%, d%, e%, f%, and g%, respectively. h is the equation error. Take the logarithm of both sides of the equation with base e, and the formula is as follows:ln(I) = ln(a) + bln(P) + cln(A) + dln(T) + eln(G) + fln(V) + gln(F) + lnh

#### 3.2.3. Scenario Analysis

Scenario analysis, a quantitative forecasting technique, examines system trajectories through socioeconomic multidimensional frameworks. This study integrates scenario analysis with the STIRPAT model to project agricultural carbon emissions in the Three Gorges Reservoir (TGRA). We calibrated driving factor variation rates using provincial policy documents, district-level population plans, agricultural census data, and historical emission trends, establishing low-, baseline-, and high-development scenarios. These scenarios identify optimal peaking pathways for carbon emissions across TGRA’s 18 districts/counties (Table 4), providing actionable insights for regional carbon mitigation strategies.

## 4. Results

### 4.1. Temporal and Spatial Distribution of GHG from Agricultural Sources in the Three Gorges Reservoir

#### 4.1.1. Time Trend of GHG Emissions from Agricultural Sources in the Three Gorges Reservoir

CH_4_ emissions from agricultural sources in the Three Gorges Reservoir ranged from 106,400 to 119,900 tons (Figure 3). The multi-year average was 113,100 tons, peaking at 119,900 tons in 2015. The multi-year average stood at 116,300 tons, peaking at 127,800 tons in 2018 with an overall pattern of gradual growth followed by decline. Concurrently, agricultural N_2_O emissions displayed a narrower range of 13,000 to 14,400 tons during the same period, maintaining a multi-year average of 13,900 tons. The peak emission level reached 14,500 tons in 2013, followed by a sustained downward trajectory. Using 100-year GWP metrics, CO_2_-equivalent emissions in the reservoir spanned 6.76–7.49 Mt (avg.7.22 Mt/yr), exhibiting initial growth then decline. This quantification reveals agriculture-climate dynamics in this ecologically critical zone.

#### 4.1.2. Spatial Emission Trend of GHG from Agricultural Sources in the Three Gorges Reservoir

(1)Spatial Distribution Patterns of Agricultural CH_4_ Emissions

The Three Gorges Reservoir exhibits pronounced spatially heterogeneous patterns in agricultural GHG emissions. Methane (CH_4_) emission hotspots (Figure 4) demonstrate significant spatial autocorrelation (annual Moran’s I = 0.42, *p* < 0.01), predominantly clustered in the reservoir’s head and central zones. Municipal-level differential analysis (*p* < 0.01) reveals marked emission reductions in urban cores (Chongqing metropolitan area, Yichang city) and Shizhu County over 15 years, contrasting with a persistent growth trajectory observed in seven riparian agricultural counties (Fuling-Wuxi corridor). Spatiotemporal evolution analysis (Figure 5) identifies a tri-phase migration of CH_4_ emission centroids: northeastward orientation, westward shift (2013 inflection point), and subsequent southwestward drift (2018 pivotal transition). This spatial polarization phenomenon elucidates the dynamic interplay between agricultural intensification and ecological conservation processes. The quantified migration pathways provide geospatial evidence for developing basin-scale zoned governance strategies, effectively bridging agroecological management with climate mitigation objectives.

(2)Spatial Distribution Patterns of Agricultural N_2_O Emissions

Spatial heterogeneity analysis at the municipal level (*p* < 0.01) reveals pronounced spatial heterogeneity in agricultural N_2_O emissions across the Three Gorges Reservoir. Riparian agricultural counties (e.g., Jiangjin, Fengdu) demonstrate persistent upward trajectories, while metropolitan cores (Chongqing, Yichang) exhibit marked downward trends (Figure 6). Spatial autocorrelation assessment indicates emissions exhibit no significant spatial clustering pattern (Moran’s I ≈ 0), presenting stochastic dispersal distribution characteristics. Spatiotemporal trajectory analysis identifies a three-phase migration pathway of N_2_O emission centroids: northeast orientation → westward shift → southwest drift (Figure 7). The western migration inflection point (2013) and southwestern transitional node (2018) constitute critical spatiotemporal milestones. This unique directional migration without spatial aggregation elucidates the dynamic spatiotemporal interplay between agricultural nitrogen management practices and watershed ecological conservation initiatives.

### 4.2. GHG Composition and Driving Factors of Agricultural Sources in the Three Gorges Reservoir

#### 4.2.1. Analysis of the Composition of GHG Emissions from Agricultural Sources in the Three Gorges Reservoir

(1)Composition Analysis of CH_4_ Emissions

Agricultural CH_4_ emissions in the Three Gorges Reservoir demonstrate distinct sectoral patterns (Figure 8). Rice cultivation remained the dominant source (52,900 tons, 49.66%), followed by livestock enteric fermentation (45,000 tons, 42.16%) and manure management (8700 tons, 8.17%). Sectoral trends revealed the following: rice-related emissions decreased progressively, livestock fermentation emissions peaked then declined, and manure emissions maintained low yet steadily decreasing proportions since 2013.

(2)Composition Analysis of N_2_O Emissions

Agricultural N_2_O emissions in the Three Gorges Reservoir demonstrate stable sectoral proportions over 15 years (Figure 9). Synthetic fertilizers constitute the dominant contributor (5900 tons, 45.08%), followed by livestock manure management (3200 tons, 24.36%). Other sources include atmospheric deposition (15.21%), and field residues (12.50%) showed minimal variation. Sectoral trends reveal the following: fertilizer-derived emissions peaked then declined, while livestock/poultry and crop residue emissions decreased progressively. Atmospheric nitrogen deposition maintained stable fluxes without distinct temporal patterns.

#### 4.2.2. Analysis of the Composition of Emission Sources at the County Level in the Three Gorges Reservoir

This study identifies critical emission hotspots in the Three Gorges Reservoir through comparative analysis of municipal versus regional emission growth rates (Figure 10). For agricultural methane (CH_4_), Chongqing’s urban core and Jiangjin District demonstrate severe methane overshoot in rice cultivation (growth exceeding 100%), necessitating prioritized paddy water management. Concurrently, districts with over 80% growth in livestock enteric fermentation emissions require feed optimization interventions. Manure management hotspots in Chongqing’s urban core and Kaizhou District (>50% growth) urgently require infrastructure upgrades, while Jiangjin, Wanzhou, Fengdu, Kaizhou, and Yunyang emerge as cross-sectoral CH_4_ priority zones.

Regarding agricultural nitrous oxide (N_2_O), fertilizer application hotspots in Fuling, Jiangjin, Kaizhou, Wulong, and Yichang (47% increase) demand precision agricultural practices. Simultaneously, livestock manure emission concerns in Chongqing’s urban core, Jiangjin, and Kaizhou necessitate improved storage systems. Straw-derived N_2_O emissions peak in Chongqing’s urban core (90.5% growth), with Fuling, Jiangjin, Zhongxian, and Kaizhou requiring enhanced straw utilization efficiency. Integrated mitigation efforts should concentrate on Chongqing’s urban core, Jiangjin, Fuling, Kaizhou, and Yichang, where overlapping CH_4_/N_2_O emission pressures reveal systemic agricultural emission challenges requiring coordinated multi-pollutant governance strategies.

### 4.3. Driving Factors of GHG from Agricultural Sources in the Three Gorges Reservoir

This study constructs a STIRPAT model for the Three Gorges Reservoir using six key variables: total population, GDP per capita, total agricultural machinery power, primary industry value-added, annual grain yield, and rural workforce. The rate of change is shown in Table 5. The variable selection rationale integrates regional characteristics as follows:

(1) Population: the population variable (7.91 million in Chongqing’s urban core) demonstrates intensified human-land conflicts. Population growth drives agricultural expansion and elevates food demand, potentially increasing GHG emissions through scaled farming operations.

(2) GDP Per Capita: the affluence indicator reveals polarized development—higher in upper/lower reservoir zones than mid-section. Economic growth correlates with agricultural modernization (mechanization and agrochemical inputs), creating complex GHG emission trade-offs.

(3) Total Power of Agricultural Machinery: the technology proxy shows spatial heterogeneity. The Chongqing urban core, Yichang, and Fuling exhibit superior mechanization. Advanced machinery reduces fossil fuel dependence, while outdated equipment in peripheral counties increases carbon intensity.

(4) Value-Added in the Primary Industry: the agricultural vitality metric highlights Jiangjin and Wanzhou as production hubs. Sectoral growth amplifies cultivation intensity, directly impacting GHG emissions through intensified farming practices.

(5) Grain Production: Production hotspots in Chongqing’s urban core and Jiangjin necessitate nitrogen-intensive practices (tillage, irrigation, and fertilization), creating nonlinear emission relationships.

(6) Number of People Employed in the Rural Areas: the labor distribution asymmetry influences agricultural intensity. Workforce concentration in specific counties escalates emission-generating activities through expanded operational scales. The STIRPAT model identifies the total power of agricultural machinery, grain production, rural workforce size, and GDP per capita as key drivers of CO_2_-eq emissions. Population, machinery power, grain production, rural workforce, and value-added in primary industry showed positive effects (Regression coefficients = 0.021–0.331) (Figure 11), while GDP per capita demonstrated a negative correlation (Regression coefficients = −0.066). These agricultural modernization indicators intensified energy consumption and labor inputs, driving carbon emissions. Studies suggest per capita emissions correlate with consumption patterns and industrial restructuring, where elevated consumption promotes high-carbon goods.

Ridge regression (K = 0.123) validated model robustness (*p* < 0.001, R^2^ = 0.789). The derived equation: lnI(CO_2_-eq) = −2.936 − 0.066lnA + 0.021lnP + 0.255lnT + 0.331lnF + 0.117lnV + 0.041 lnG, translates to:*I =* e^−2.936^*A*^−0.066^*P*^0.021^*T*^0.255^*F*^0.331^*V*^0.117^*G*^0.041^

### 4.4. Prediction of the Peak Trend of GHG Carbon from Agricultural Sources in the Three Gorges Reservoir

Figure 12 shows the CO_2_-eq emissions from agricultural sources in the 18 districts and counties of the Three Gorges Reservoir in the future. The CO_2_-eq emissions from agricultural sources in the 18 districts and counties of the Three Gorges Reservoir in the future are influenced by multiple factors. In most districts and counties, the peak cumulative CO_2_-eqemissions under the high-speed development scenario are higher than those under the low-speed development scenario and the baseline scenario, such as Chongshou District, Jiangjin District, Wanzhou District, Wuxi County, Zigui County, Yichang City District, etc. In a few districts and counties, such as the central urban area of Chongqing, Kaizhou District, and Badong County, the peak cumulative CO_2_-eq emissions under the low-speed scenario are higher than those under the baseline scenario and the high-speed development scenario. Although all 18 districts and counties are located within the Three Gorges Reservoir, due to the differences in population, economy and technological development among the districts and counties, they will have different development trajectories. Therefore, the peak CO_2_-eq emissions in the future will be influenced by multiple factors. It is necessary to study the reduction of emissions under different scenarios for each district and county in the future.

China’s subnational carbon peaking timelines exhibit significant spatial heterogeneity across development scenarios (Figure 13, Figure 14 and Figure 15). Under the low-speed scenario, 12 districts/counties are projected to achieve carbon neutrality by 2030, with Wulong, Wushan, and Changshou reaching peak emissions by 2035, while Fuling and Wuxi lag behind, not peaking until 2040. The Three Gorges Reservoir baseline scenario suggests 11 districts/counties will meet 2030 targets, including Zhongxian and Wushan attaining 2035 milestones, whereas Wulong, Badong, Wushan, and Changshou are anticipated to peak before 2040, contrasting with Fuling and Wuxi’s post-2040 trajectory. Notably, under high-development conditions, 10 districts/counties maintain 2030 compliance despite systemic delays, with Changshou, Zhongxian, Badong, Wushan, and Yichang achieving 2035 benchmarks, while Fuling and Wuxi cross the 2040 threshold, and Wulong persists beyond 2040. These multi-scale variations underscore the necessity for tailored mitigation strategies across the reservoir area.

China’s carbon neutrality commitment necessitates subnational implementation frameworks that transcend singular peaking timelines, particularly within the Three Gorges Reservoir’s 18 districts (2021–2045). This study establishes a spatiotemporal visualization paradigm where geospatial bubbles encode cumulative emission magnitudes through proportional sizing, chronological positioning (2030 green baseline vs. 2045+ red threshold), and scenario differentiation via chromatic coding (green: low-speed; pink: baseline; and red: rapid development). Twelve districts demonstrate pre-2030 peaking capacity across scenarios, mandating pathway optimization through comparative emission integrals—exemplified by Chongqing’s urban core where baseline conditions minimize long-term emission loads, contrasted with Yichang’s temporally divergent patterns (2025 baseline vs. pre-2021 low-speed vs. 2030 rapid development peaks). This framework treats peaking milestones as dynamic variables in multi-scale decarbonization, emphasizing context-specific optimization based on district-level development trajectories. (Figure 16; Table 6).

## 5. Discussion

(1)Spatial Patterns and Policy Implications

The spatial distribution of CH_4_ and N_2_O emissions at the county level reveals critical hotspots requiring prioritized mitigation. Agricultural GHG fluxes in the Three Gorges Reservoir show pronounced spatial–temporal variations, exemplified by Zigui County’s livestock-driven CH_4_ fluctuations (2006–2020) and Chongqing’s urban N_2_O decline from cropland conversion. These shifts correlate with policy interventions like rural revitalization initiatives and the Five-Year Plans for livestock pollution control. Targeting spatial clusters with elevated emission levels enables optimized agricultural zoning, enhanced interregional collaboration, and scenario-specific mitigation protocols. Notably, the area’s 2018 agricultural CO_2_-eq intensity (0.015% of national emissions per 0.011% land area) [49] underscores the necessity for precision management in topographically complex, agriculturally intensive zones.

(2)Sectoral Contributions and Mitigation Pathways

Rice cultivation constitutes the primary CH_4_ source (48–52% of agricultural emissions), particularly in reservoir-tail regions like Chongqing and Fuling, where expanded paddy fields increased emissions by 18% (2017–2020). Livestock operations in central reservoir zones (Wanzhou, Kaizhou) contribute 63–67% of N_2_O emissions through manure management and fertilizer application. Despite stable emission patterns over 15 years, mechanization constraints and escalating grain demands perpetuate chemical fertilizer reliance in mountainous counties (Yunyang, Wanzhou) [50]. Strategic interventions should prioritize: (1) organic fertilizer substitution in high-population croplands; (2) advanced livestock feed optimization systems; and (3) terraced farming adaptations to reduce machinery dependency. These measures are tailored to the reservoir’s distinct agroecological characteristics and target emission-intensive sectors effectively.

(3)Agricultural Carbon Dynamics in the Three Gorges Reservoir: Multidimensional Drivers and Decarbonization Pathways

Agricultural carbon dynamics in the Three Gorges Reservoir are principally governed by socioeconomic modernization indicators—total power of agricultural machinery, grain production, and number of people employed in rural areas—which collectively demonstrate regulatory influences on CO_2_-eq emissions through mechanization-driven energy intensification and labor reorganization. While population growth and value-adding in the primary industry stimulate livestock expansion and associated GHG fluxes, paradoxical suppression effects emerge where rural workforce development preferentially fuels non-agricultural sectors, and diversified crop cultivation decouples yield growth from emission trajectories. GDP per capita exhibits dual regulatory mechanisms: high-income regions achieve emission-intensity reduction through advanced agro-technologies, whereas low-income areas face elevated per-unit emissions from traditional practices and land conversion. Urbanization-mediated population migration further introduces N_2_O mitigation through enhanced environmental awareness, though demographic shifts (aging populations and household structure changes) [51,52] present unresolved climate governance challenges. This complex interplay necessitates integrated analytical frameworks that reconcile agricultural modernization rates, consumption pattern evolution, and institutional policy architectures to optimize context-specific decarbonization pathways.

(4)Scenario-Driven Disparities in Agricultural Emission Peaks: Predictive Complexities and Policy Imperatives

The peak value of cumulative CO_2_-eq emissions under the high scenario is usually higher than that in the low scenario and the baseline scenario, which may be due to the following reasons: under the high scenario, due to the increase in the power of agricultural machinery and the primary output value, the production level of agriculture may increase, and the expansion of large areas of arable land or the reduction of forests may lead to an increase in the use of chemical fertilizers, pesticides, and other chemicals in agricultural activities, thereby increasing emissions. It is worth the attention of scholars that, for a district and county, because the prediction is evidence of the complexity of the potential emission trajectory and peak path in the future, these factors may change in the future, and researchers can warn and provide theoretical direction for future policies, and the final result also needs the guidance of district and county policies, the innovation of low-carbon technology, and the improvement of people’s environmental awareness to help achieve the carbon peak as soon as possible. It is worth studying that for a district and county, the carbon emissions under the low scenario are not necessarily greater than those under the high and high scenarios, because the positive and negative factors under the low and high scenarios are different at the same time. This can be explained as evidence of the complexity of potential emission trajectories and peak pathways, and most studies at home and abroad have used regression models to predict the timing of carbon peaking in other regions of China, and the prediction of emission trajectories in China’s provinces, especially peak emissions, has been an area of increasing concern. Figure 17 shows that other scholars have predicted the predicted peak time of seven provinces in China based on different regression models, and this study compares it with other studies [53,54,55,56,57,58,59,60,61,62]. For Chongqing, the peak estimation in this study is generally consistent with the peak time of others, and most of the other studies have proved that the peak time is concentrated in 2030 and 2035, and there are similar studies with similar results, such as Shandong Province, which is expected to reach the peak time in 2045. Many scholars have studied different aspects and different areas of carbon emission peaking time, which further reminds people of the urgency of the Chinese government to achieve carbon peak before 2030.

## 6. Conclusions

In this study, the Three Gorges Reservoir of China, a special area with single agricultural structure and high-intensity chemical fertilizer application, was used to estimate and analyze the emissions of GHG methane (CH_4_) and nitrous oxide (N_2_O) from agricultural sources in the Three Gorges Reservoir in China from 2006 to 2020 using the method recommended by the Intergovernmental Panel on Climate Change (IPCC) and 18 county-level units in the Three Gorges Reservoir as the research scale. The STIRPAT model was used to analyze the driving factors behind the GHG emissions from agricultural sources, and three different social development scenarios were constructed to explore the carbon peak time of GHG from agricultural sources in 18 districts and counties in the Three Gorges Reservoir in the future.

(1)Spatiotemporal Characteristics of GHG Emissions

From 2006 to 2020, agricultural methane (CH_4_) emissions in the Three Gorges Reservoir remained between 106.4 and 119.9 thousand tons, with a spatial distribution showing a clustered pattern of higher emissions in the southwest and lower emissions in the northeast, accompanied by significant positive spatial autocorrelation. The emission centroid shifted sequentially from the southwest to the northeast and back to the southwest. Nitrous oxide (N_2_O) emissions fluctuated between 13.0 and 14.4 thousand tons, exhibiting random spatial distribution with a southwestward migration of the centroid. The 100-year Global Warming Potential (GWP) of CO_2_-equivalent emissions ranged from 6.76 to 7.49 million tons, displaying an initial increase followed by a gradual decline, with CH_4_ demonstrating notable spatial aggregation.

(2)Emission Source Composition

Rice cultivation dominated CH_4_ emissions (50%), though emissions decreased overall; exceptions included Chongqing’s central urban area and Jiangjin District, where emissions surged by over 100%. Livestock (enteric fermentation + manure management) constituted the secondary CH_4_ source. Fertilizer application contributed 45% of N_2_O emissions, with Fuling, Jiangjin, Kaizhou, Wulong, and Yichang urban districts experiencing growth rates exceeding 47%. Livestock manure accounted for 24% of N_2_O emissions, with both sources showing an initial rise followed by reduction.

(3)Driving Mechanism Analysis

The STIRPAT model identified agricultural machinery power, grain yield, rural workforce, and primary industry output as key positive drivers of CO_2_-eq emissions, with machinery power and grain yield exhibiting the strongest effects. In contrast, per capita GDP exerted inhibitory effects through technological and managerial advancements, revealing an inverse relationship between economic development and emission intensity.

(4)Multi-Scenario Peaking Projections

Under low-, baseline-, and rapid-development scenarios, 12, 11, and 10 districts/counties, respectively, are projected to achieve carbon peaking before 2030. Five regions—Chongqing’s central urban area, Fuling, Jiangjin, Wanzhou, and Kaizhou—consistently exhibit peaking volumes exceeding 500,000 tons CO_2_-eq across all scenarios, necessitating prioritized monitoring. Regions should implement tailored pathways through policy interventions, low-carbon innovation, and public engagement based on regional development contexts to accelerate peaking targets.

## Figures and Tables

**Figure 1 microorganisms-13-01217-f001:**
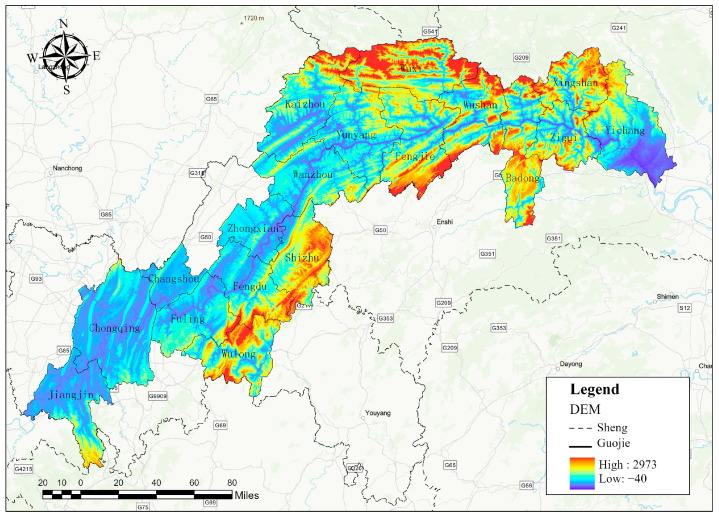
Location map of the Three Gorges Reservoir.

**Figure 2 microorganisms-13-01217-f002:**
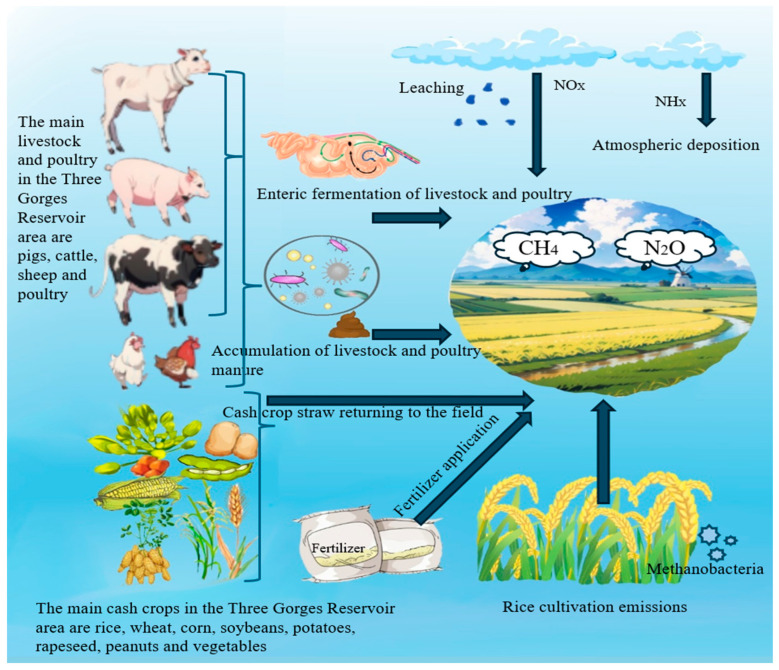
Sources of GHG emissions from agricultural activities.

**Figure 3 microorganisms-13-01217-f003:**
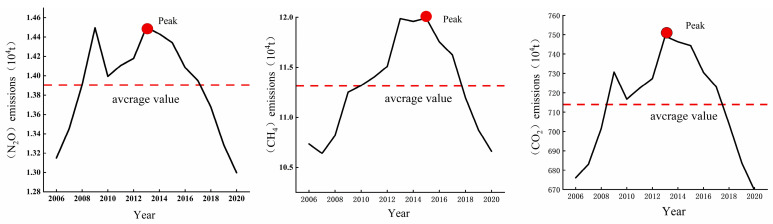
Total emissions of CH_4_, N_2_O, and CO_2_-eq from agricultural sources in the Three Gorges Reservoir.

**Figure 4 microorganisms-13-01217-f004:**
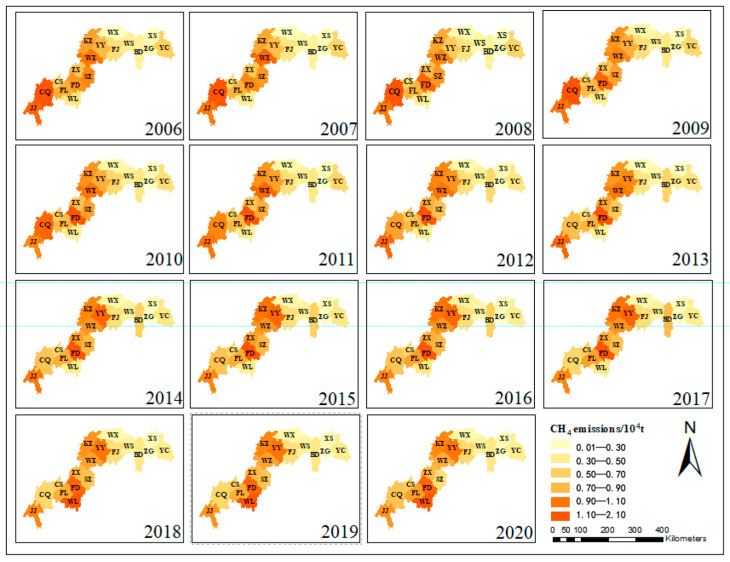
Spatial distribution of CH_4_ from 2006 to 2020.

**Figure 5 microorganisms-13-01217-f005:**
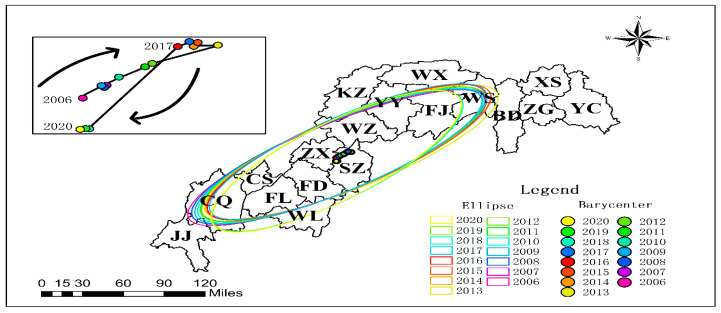
CH_4_ center of gravity trajectory migration diagram.

**Figure 6 microorganisms-13-01217-f006:**
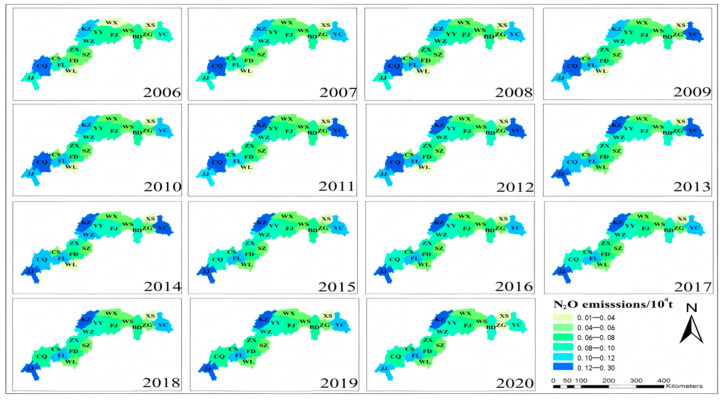
Spatial distribution of N_2_O from 2006 to 2020.

**Figure 7 microorganisms-13-01217-f007:**
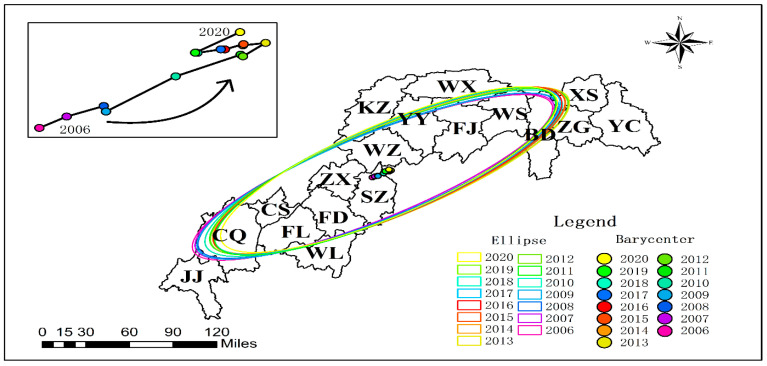
N_2_O center of gravity trajectory migration diagram.

**Figure 8 microorganisms-13-01217-f008:**
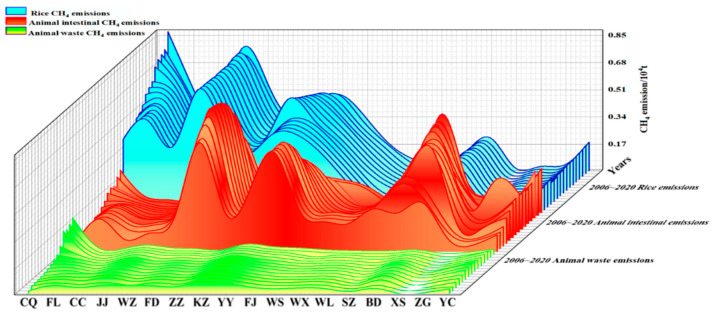
Waterfall diagram of the emission trend of CH_4_ emission sources from agricultural sources in the Three Gorges Reservoir in the past 15 years.

**Figure 9 microorganisms-13-01217-f009:**
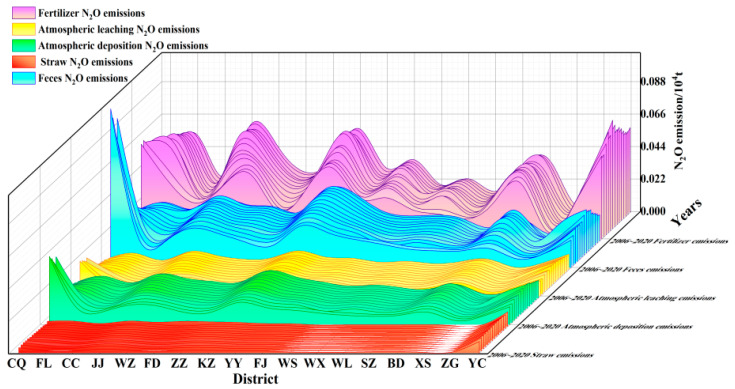
Waterfall diagram of the emission trend of N_2_O emission sources from agricultural sources in the Three Gorges Reservoir in the past 15 years.

**Figure 10 microorganisms-13-01217-f010:**
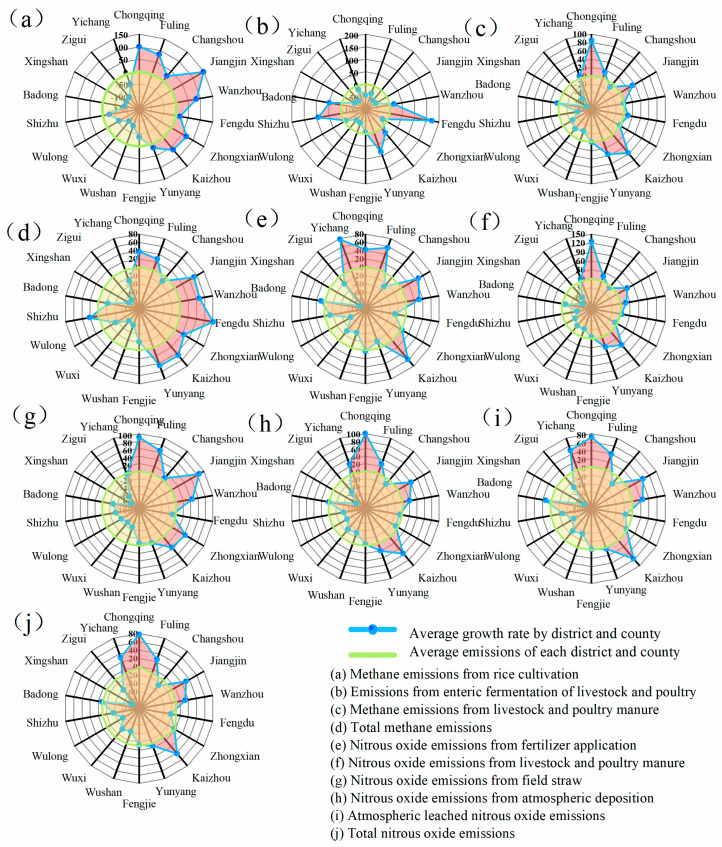
Analysis of the composition of N_2_O and CH_4_ in agricultural sources at the county level in the Three Gorges Reservoir.

**Figure 11 microorganisms-13-01217-f011:**
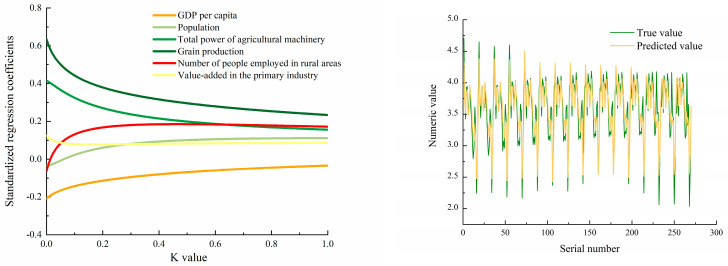
Ridge regression plots and model prediction plots for CO_2_-equivalent (CO_2_-eq) emission.

**Figure 12 microorganisms-13-01217-f012:**
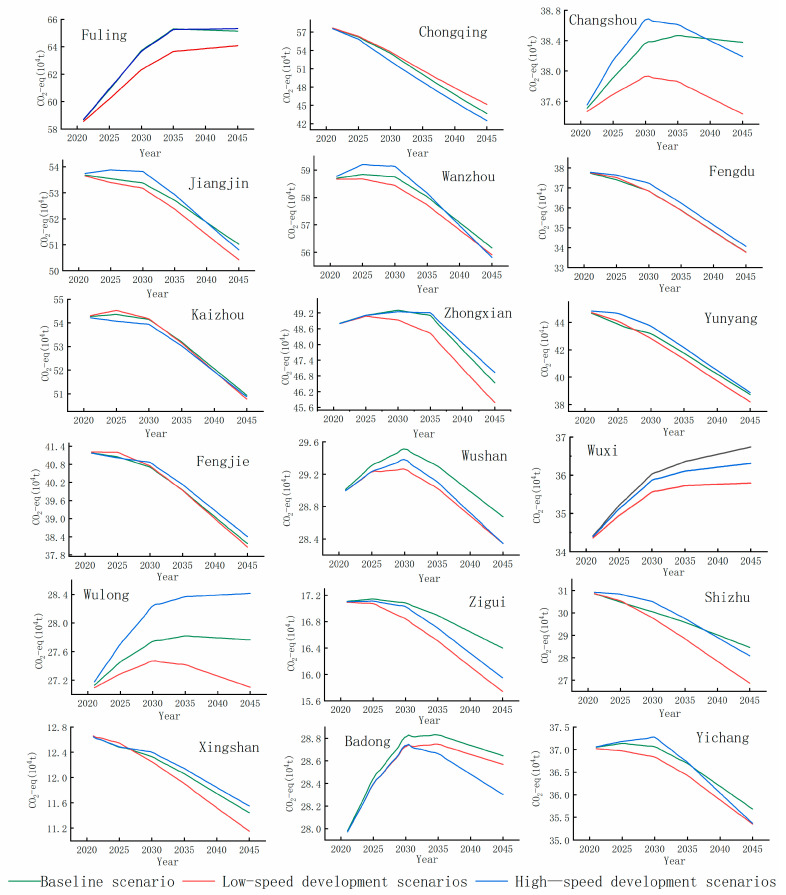
Carbon peaking trend under three scenarios in each district and county of the Three Gorges Reservoir.

**Figure 13 microorganisms-13-01217-f013:**
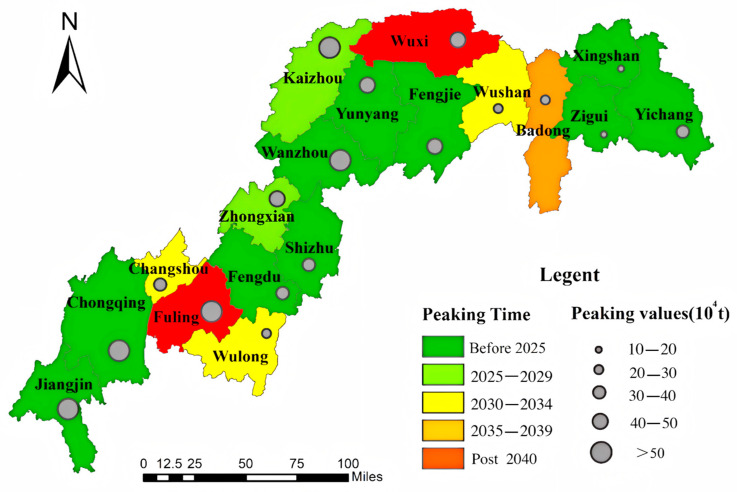
Carbon peaking timing and magnitude under low-speed development scenario.

**Figure 14 microorganisms-13-01217-f014:**
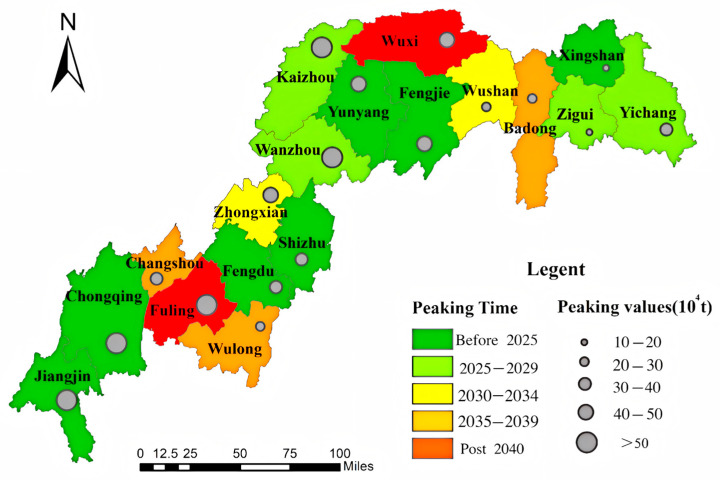
Carbon peak projections in baseline scenario.

**Figure 15 microorganisms-13-01217-f015:**
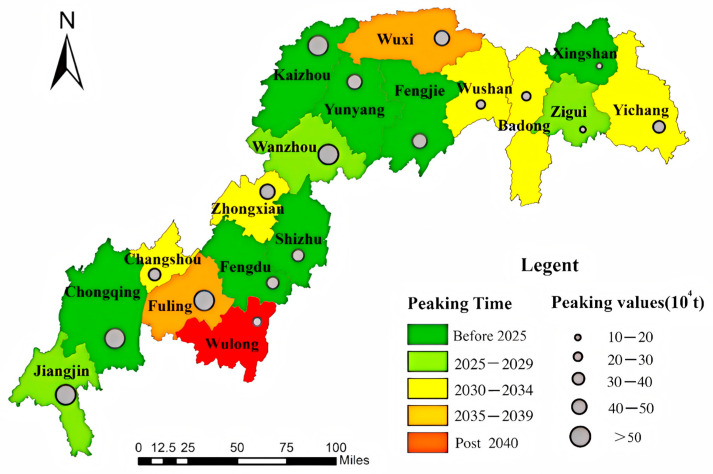
Carbon peaking dynamics under rapid development scenario.

**Figure 16 microorganisms-13-01217-f016:**
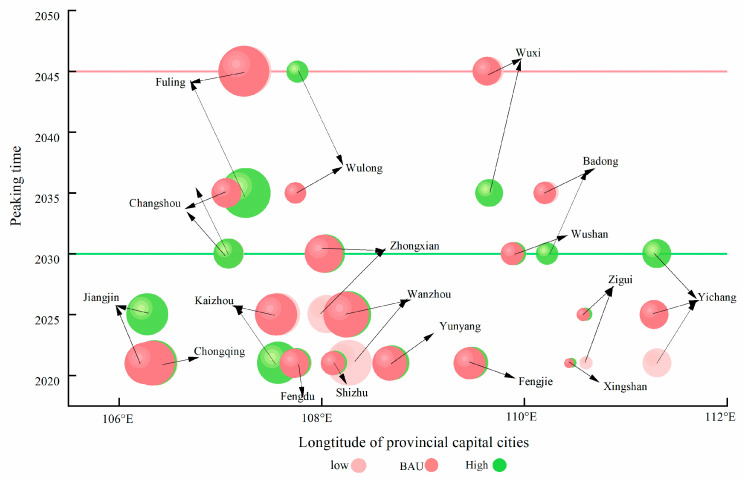
The optimal carbon peak pathway for each district and county in the Three Gorges Reservoir.

**Figure 17 microorganisms-13-01217-f017:**
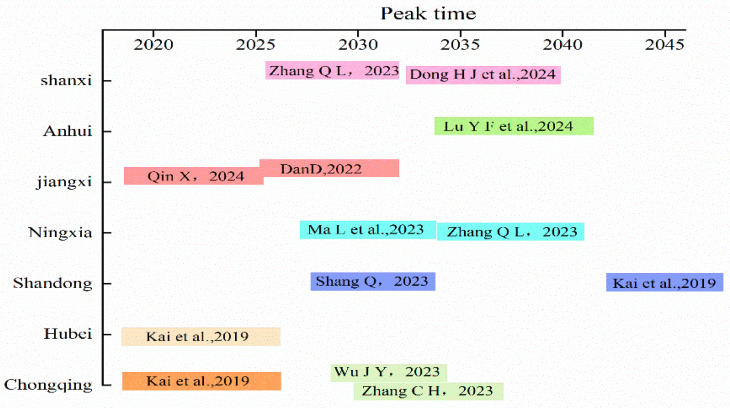
Similar studies on carbon peak times.

**Table 1 microorganisms-13-01217-t001:** Main animal nitrogen excretion and emission factor indicators.

Animals (Southwest) (kg/Head/Year)	Cattle	Poultry	Goat	Pig
Nitrogen excretion	40	0.6	12	16
Animal intestinal CH_4_ emission factor	69.9	—	3.6	1
CH_4_ fecal emission factor	1.953	0.012	0.133	0.770
N_2_O fecal emission factor	1.197	0.007	0.064	0.159

**Table 2 microorganisms-13-01217-t002:** Main crop parameters.

Crop	Dry-to-Weight Ratio	Nitrogen Content of Straw	Economic Coefficients	Root-to-Crown Ratio
rice	0.885	0.00753	0.489	0.125
wheat	0.87	0.00516	0.434	0.166
corn	0.86	0.0058	0.438	0.17
soybean	0.86	0.0181	0.425	0.13
potato	0.45	0.011	0.667	0.05
rapeseed	0.82	0.00548	0.271	0.15
peanut	0.90	0.0182	0.556	0.2
vegetable	0.15	0.008	0.83	0.25

**Table 3 microorganisms-13-01217-t003:** The meaning of each variable in the model.

Variable	Symbol	Meaning	Unit
GHG emissions	I	CH_4_ or N_2_O emissions in the Three Gorges Reservoir	Tons of tons
Population size	p	Total population	10^4^ people
Affluence	A	Gross National Product per capita	yuan
Technology	T	Total power of agricultural machinery	vandrite
The vitality of agricultural development	G	Primary industry-added value	10^4^ yuan
Grain production	F	Annual grain production	ton
Labor vitality in the reservoir area	V	Number of rural employees	people unit

**Table 4 microorganisms-13-01217-t004:** Ridge regression results analysis chart.

K = 0.123	Regression Coefficients	Standard Error	*T*-Test Value	*p*-Value	R^2^
Constant	−2.936	0.349	−8.404	0.000 ***	0.789
GDP per capita	−0.066	0.014	−4.553	0.000 ***	
Population	0.021	0.022	0.951	0.342	
Total power of agricultural machinery	0.255	0.029	8.694	0.000 ***	
Grain production	0.331	0.025	13.375	0.000 ***	
Number of people employed in the rural areas	0.117	0.025	4.632	0.000 ***	
Value-added in the primary industry	0.041	0.018	2.348	0.020 **	

** means significant at 95% confidence level and *** means significant at 99% confidence level.

**Table 5 microorganisms-13-01217-t005:** District and county scene setting (taking Chongqing urban area from 2020 to 2025 as an example).

District	Drivers	Benchmark Scenario	Low-Speed Development Scenario	High Speed Development Scenario
		(%)		
Chongqing	GDP per capita	8	6	10
	Population	1.8	1.4	2.2
	Total power of agricultural machinery	0.6	0.3	1
	Grain production	−0.8	−0.6	−1
	Number of people employed in the rural areas	−3	−2.5	−4
	Value-added in the primary industry	8	5	10

**Table 6 microorganisms-13-01217-t006:** Optimal pathways for carbon peaking in 18 districts and counties in the Three Gorges Reservoir.

Scenario	Baseline Scenario	Low-Speed Development Scenarios	Rapid Development Scenarios
District	Chongqing, Jiangjin, Fengdu, Yunyang, Fengjie	Changshou, Wanzhou, Zhongxian, Wushan, Wulong, Shizhu, Xingshan, Yichang, Zigui	Fuling, Badong, Kaizhou, Wuxi

## Data Availability

Data will be made available on request.

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
