# Peer review of "How Long Until Agricultural Carbon Peaks in the Three Gorges Reservoir? Insights from 18 Districts and Counties"

_microorganisms, 2025, doi:10.3390/microorganisms13061217_

Round 1
Reviewer 1 Report
Comments and Suggestions for Authors
I read a very interesting study that is needed in the international literature from the point of view of ongoing climate change and increasing greenhouse gas emissions. The subject of the study is the measurement of greenhouse gas emissions associated with agriculture in areas with intensive agriculture in China. These are areas with extremely intensive production, reinforcing the need for this type of research. The strength of this research is the coverage of a large area and the long measurement period. The research method was rightly used, based on the IPCC methodology, which I consider necessary. Good use was made of the method for estimating emissions. The methods used can be well applied to studies in other parts of the world. The authors fill a scientific gap in this area with their research. The structure of the study and the visualisation is acceptable. However, there is a need for an additional theoretical chapter (after the introduction). Such an additional chapter is necessary because a good international study should have a solid review of the world literature. I would ask the authors to make a solid review of the literature on agricultural emissions in other parts of the world and the projection of these emissions in this chapter. Much of this type of research is being done in the EU. Please review the EU's climate neutrality strategy to 2050. Please use the world literature extensively. The study will then take on even greater quality. I appreciate the discussion of the results, it must also be supported by results from other parts of the world. I value the idea of the study itself highly.
Author Response
Comment 1 needs an extra theoretical chapter (after the introduction). Such an extra chapter is necessary, because a good international study should have a solid review of world literature. I would like to ask the author to make a solid review of the agricultural emission literature in other parts of the world and the prediction of these emissions in this chapter. Many such studies are conducted in the European Union. Please refer to the EU's climate neutrality strategy to 2050. Please make extensive use of world literature. Then, the research will be of higher quality.
Response 1 Thank you for pointing out this problem. I have reorganized the domestic and foreign literatures, and listed a new chapter 2. Global agricultural emissions: models, projections and policy frameworks after the introduction, summarizing the current research on agricultural greenhouse gases from two aspects, reorganizing the introduction and introducing other references to compare with this study.
Reviewer 2 Report
Comments and Suggestions for Authors
The authors set the stage to estimate and analyze the emissions of greenhouse gases (GHGs) methane and nitrous oxide from agricultural sources in the Three Gorges Reservoir in China from 2006- 2020 using the method recommended by the Intergovernmental Panel on Climate Change (IPCC) and 18 county-level units in the Three Gorges Reservoir as the research scale. The STRIRPAT model was used to analyse the driving factors behind the GHG emissions from agricultural sources, and three different social development scenarios were constructed to explore the carbon peak of the GHG from agricultural sources. The results revealed that the methane emissions in Three Gorges Reservoir remained between 106.4–119.9 thousand tons, with a spatial distribution showing a clustered pattern of higher emissions in the southwest and lower emissions in the northeast. Nitrous oxide (N2O) emissions fluctuated between 13.0 – 14.4 thousand tons, exhibiting random spatial distribution with a southwestward migration of the centroid. Rice cultivation dominated methane emissions (50%), though emissions decreased overall. Livestock (enteric fermentation + manure management) constituted the secondary methane source, whereas fertilizer application contributed 45% of N2O emissions. Livestock manure accounted for 24% of N2O emissions, both sources showing an initial rise followed by reduction. The STIRPAT model identified agricultural machinery power, grain yield, rural workforce, and primary industry output as key positive drivers of CO2-eq emissions, with machinery power and grain yield exhibiting the strongest effects. In addition, under low-, baseline-, and rapid-development scenarios, 12, 11, and 10 districts/counties, respectively, are projected to achieve carbon peaking before 2030. The derived decision-support system provides empirically grounded solutions for aligning subnational climate actions with global mitigation targets. The authors made an interesting analysis of the GHGs emissions in the Three Gorges Reservoir in China, however there are some issues that have to be addressed.
1) In introduction, the authors should mention if there are other studies of spatiotemporal distribution of GHGs and sources of carbon emission in other regions of China and compare their results with their own study. 2) The equations describing the calculation of methane, N2O emissions should be numbered. 3) a) Figures of spatial distribution of methane and N2O from 2006 - 2020 and b) waterfall diagrams of methane and N2O emissions should be presented. 4) A spider diagram (similar of figure 10) for CO2 emissions at the country level in the Three Gorges Reservoir should be presented and analysed.Author Response
Comment 1
1) In the introduction, the author should mention whether there are other studies on the temporal and spatial distribution of GHG and the sources of carbon emissions in other parts of China, and compare their findings with their own. 2) Equation 2O emissions describing the calculation of methane n shall be numbered. 3) a) Spatial distribution map of methane and N2O from 2006 to 2020 and b) Waterfall map of methane and N2O emission should be provided. 4) The spider diagram of CO (similar to Figure 10)2 should introduce and analyze the emissions of the Three Gorges Reservoir at the national level. Response 1
Thank you for pointing out the problem, which is of great help to my research. I have reorganized the domestic and foreign literature and listed a new chapter after the introduction. 2. Global agricultural emissions: models, projections and policy frameworks, summarizing the current research on agricultural greenhouse gases from two aspects. In view of your research on the temporal and spatial distribution of GHG and the sources of carbon emissions in other areas of China, and comparing their research results with my own research, I have rearranged the introduction and introduced other references to compare with this research. The equations introduced in this paper have also been renumbered. The waterfall maps of methane and nitrous oxide are made in Figure 8 and Figure 9 according to the year. As for the emissions at the national level in the Three Gorges reservoir area you mentioned, I have added a comparison in the quotation.
Round 2
Reviewer 1 Report
Comments and Suggestions for Authors
I thank the authors for significantly improving the study. Thank you that they added a theoretical chapter, which clearly improved the quality of the study, it is an important introducing chapter to the analyzed subject matter. The asset is undoubtedly the well-applied method of analysis, visualization and interesting discussion of the results. I consider that all my comments have been taken into account. The study is properly structured. The topic is very topical, showing how greenhouse gas emissions are shaped in areas with intensive agriculture. The study will certainly be widely cited.
Reviewer 2 Report
Comments and Suggestions for Authors
-